# Prevalence of Arterial Hypertension and Characteristics of Nocturnal Blood Pressure Profile of Asthma Patients According to Therapy and Severity of the Disease: The BADA Study

**DOI:** 10.3390/ijerph17186925

**Published:** 2020-09-22

**Authors:** Domenico Di Raimondo, Gaia Musiari, Alida Benfante, Salvatore Battaglia, Giuliana Rizzo, Antonino Tuttolomondo, Nicola Scichilone, Antonio Pinto

**Affiliations:** 1Division of Internal Medicine and Stroke Care, Department of Promoting Health, Maternal-Infant, Excellence and Internal and Specialized Medicine (Promise) G. D’Alessandro, University of Palermo, 90133 Palermo, Italy; gaiamusiari@gmail.com (G.M.); giulianarizzo@yahoo.it (G.R.); bruno.tuttolomondo@unipa.it (A.T.); antonio.pinto@unipa.it (A.P.); 2Division of Respiratory Diseases, Department of Promoting Health, Maternal-Infant, Excellence and Internal and Specialized Medicine (Promise) G. D’Alessandro, University of Palermo, 90133 Palermo, Italy; benfantealida@gmail.com (A.B.); salvatore.battaglia@unipa.it (S.B.); nicola.scichilone@unipa.it (N.S.)

**Keywords:** asthma, cardiovascular risk, essential hypertension, inhaled corticosteroid therapy, nocturnal dipping of blood pressure

## Abstract

Background: several studies report an increased risk for asthmatic subjects to develop arterial hypertension and the relationship between these two diseases, frequently co-existing, still has some unclear aspects. Methods: The BADA (blood pressure levels, clinical features and markers of subclinical cardiovascular damage of asthma patients) study is aimed to evaluate the prevalence of the cardiovascular comorbidities of asthma and their impact on the clinical outcome. The main exclusion criteria were the presence of other respiratory diseases, current smoking, any contraindication to ambulatory blood pressure monitoring (ABPM). Results: The overall percentage of asthmatics having also hypertension was 75% (30 patients) vs. 45% (18 patients) of the control group (*p*: 0.012). Reduced level of FEV_1_ (but not inhaled steroid therapy) was associated to newly-diagnosed hypertension (*p*: 0.0002), higher day SBP levels (*p*: 0.003), higher day DBP levels (*p*: 0.03), higher 24 h-SBP levels (*p*: 0.005) and higher 24h-DBP levels (*p*: 0.03). The regression analysis performed taking into account sex, age, diabetes, fasting glucose, and body mass index confirms the independent role played by asthma: odds ratio (OR): 3.66 (CI: 1.29–11.1). Conclusions: hypertension is highly prevalent in asthma; the use of ABPM has allowed the detection of a considerable number of unrecognized hypertensives.

## 1. Introduction

Asthma is one of the most common chronic, non-communicable diseases, affecting around 334 million people worldwide [1]. The prevalence of asthma increased progressively over the latter part of the last century and is still increasing with a predicted additional 100 million people affected by 2025 [2]. The global prevalence of self-reported, doctor-diagnosed asthma in adults is 4.3%, with wide variation between countries [3]. Similarly, essential hypertension is one of the leading chronic diseases worldwide, with an overall prevalence in adults that is around 30–45% [4,5], and a global age standardized prevalence of 24 and 20% in men and women, respectively, in 2015 [5,6]. Increased rates of hypertension in asthma patients are observed in most of the studied populations independent of traditional risk factors [7,8,9]. Dogra et al. [8] reports that asthmatics were 43% (OR: 1.43, CI 1.19–1.72) more likely to have heart disease, and 36% (OR: 1.36, CI 1.21–1.53) more likely to have high blood pressure than non-asthmatics. Similarly, Zhang et al. [9], in a large sample of Canadian subjects (over 130,000 subjects), found a 39% increased risk (OR: 1.39, CI 1.27–1.53) for hypertension. A study conducted among Arab Americans reveals a closer association between asthma and hypertension (OR: 2.3, CI 1.30–4.22) [10]; furthermore, Panek et al., recently showed that the prevalence of hypertension in asthma increases with the severity of the disease: 22.9% (mild) vs. 29.3% (moderate) and 35.7% (severe); the control group had a 21% lower probability of being hypertensive (OR: 0.79, CI 0.48–1.3) [11]. None of these studies evaluated the event of an unrecognized hypertension and/or the 24-h blood pressure profile of asthmatic patients through an ambulatory blood pressure monitoring (ABPM).

The reasons of the association between asthma and high blood pressure are to large extent unknown. Of note, asthma medications may play a role over development of hypertension: long acting bronchodilators (LABA) have been associated with mortality in asthma populations [12]. Systemic corticosteroids have been shown to have undesirable effects, such as hypertension and other adverse cardiovascular outcomes [13,14]. Inhaled corticosteroids (ICS) at higher doses are absorbed systemically [15]. Several authors raise the possibility that corticosteroid treatment (both oral and inhaled) may promote the development of hypertension in genetically predisposed asthmatics by mineral corticosteroid effects, with retention of sodium and fluid [13,14]. In this regard, an interesting hypothesis is put forward by Ferguson et al. [16], who suggests a “double-edged sword” relationship of ICS with hypertension: lower doses of ICS may confer a “protective” association while the opposite may be applicable for higher doses.

The BADA (blood pressure levels, clinical features and markers of subclinical cardiovascular damage of asthma patients) study aimed to evaluate the prevalence of the cardiovascular comorbidities of asthma and their impact on the course of the disease. The main objectives of this pilot study were to evaluate the prevalence of hypertension in asthmatics adding to the cases with already known hypertension those identified by the execution of a 24-h (h) ambulatory blood pressure monitoring (ABPM). This type of approach, which also provides additional information concerning, for example, the circadian rhythm of blood pressure sets the basis for a more in-depth evaluation of the factors influencing the occurrence of arterial hypertension in asthmatic patients, which is the future objective of the BADA Study.

## 2. Materials and Methods

In this pilot study forty patients affected by asthma were consecutively recruited among those referring to outpatient clinic of the Division of Respiratory Diseases of the University of Palermo, Italy, in the period between 01/01/2018 and 31/03/2019. All subjects enrolled were Caucasian, were previously informed about the characteristics and the object of the study and a written informed consent was obtained. The study was approved by the ethical committee of our institution (prot 018/2017).

Exclusion criteria:-Diagnosis of secondary hypertension-Co-existing chronic obstructive pulmonary disease (COPD)-Co-existing obstructive sleep apnea (OSA)-Current smoking. Current smoking was defined based on a positive response to the question “Do you currently smoke any tobacco products such as cigarettes, cigars, or pipes?”-A medical history of an acute vascular event (ischemic or hemorrhagic stroke, acute myocardial infarction, acute limb ischemia) in the previous six months.-Every condition contraindicating the reliability and/or the execution of the ABPM:
Finding of supraventricular arrhythmias (atrial flutter, paroxysmal, persistent or permanent atrial fibrillation);History or clinical evidence of orthostatic hypotension, evaluated comparing sitting and standing Blood Pressure (BP) during the run-in visit;Clinical history of autonomic dysfunction or diabetic neuropathy;Severe obesity; excessive alcohol consumption; history of sleep disturbance; night-workers;Intolerance to the procedure.

A total of forty consecutive individuals referring to the Division of Internal Medicine and Stroke Care of the University of Palermo for ambulatory examination due to medical reasons different from respiratory diseases served as the control group. All patients in the control group were Caucasian, should not have a history of asthma, should not take steroid therapy for any reason, and should have no evidence of airflow limitation to the spirometry.

The initial study procedure included, for all subjects, a comprehensive recording of medical history with specific attention to cardiovascular risk profile, pharmacological anamnesis, complete physical examination, assessment of body mass index (BMI), calculated as the individual’s body weight divided by the square of his or her height, and the following blood biochemical examinations (total cholesterol, HDL cholesterol, triglyceride, creatinine, fibrinogen, complete blood count, fasting glucose, C-reactive protein). Subjects were defined as type 2 diabetics if they had known diabetes treated by diet, oral hypoglycemic drugs or insulin. Previous cerebrovascular disease (TIA/ischemic stroke) was assessed by history, specific neurological examination performed by specialists, and hospital or radiological (brain computed tomography or brain magnetic resonance) records of definite previous stroke. Previous coronary heart disease was detected by history, clinical examination, electrocardiogram and echocardiogram. According to the available anamnestic information collected at the time of enrolment there were no significant differences between cases and controls regarding the characteristics of the diet and the level of physical activity (all subjects enrolled were sedentary).

### 2.1. Respiratory Evaluation

Asthmatic patients enrolled met the diagnosis of the disease based on Global Initiative for Asthma (GINA) criteria [17], considering the history of respiratory symptoms and the functional evidence of reversible bronchial obstruction on the basis of the spirometric examination. Lung function tests were performed according to European Respiratory Society/American Thoracic Society (ERS/ATS) recommendations [18]. The analysis included the following parameters: FEV_1_ (forced expiratory volume in 1 s) expressed in liters and in percent predicted value (FEV_1_%), FVC (forced vital capacity) expressed in liters and in percent predicted value (FVC%) and the FEV_1_/FVC ratio, expressed as absolute values. Reversibility of bronchial obstruction was assessed. Severe asthma was defined as asthma which requires treatment with high dose inhaled corticosteroids (ICS) plus a second controller (and/or systemic corticosteroids) or which remains uncontrolled despite this therapy [17]. The severity of the disease was assessed according to the treatment level to obtain control of asthma symptoms at the time of the first examination. The quantification of the daily intake of ICS, in relation to patient age (in mcg/day) was obtained according to the clinical comparability table indicated by GINA document [17]. In order to accomplish the analysis of the sample the asthmatic patients were stratified in accordance with the FEV_1_ value (from the highest to the lowest), dividing the sample into quartiles and comparing the top quartile (including patients with the best respiratory function) with the bottom quartile.

### 2.2. Cardiovascular Evaluation

Essential hypertension was defined based on clinical history, hospital records and/or antihypertensive treatments taken daily by the patients. For these subjects, no change of the antihypertensive drugs regularly taken was allowed during the interval of the study. A newly diagnosed hypertension after the ABPM execution was established according to the exceeding of threshold values indicated by the ESC-ESH 2018 Guidelines [5] for an ABPM-based diagnosis. ABPM was performed by a TM–2430 Recorder by A & D Company Limited of Tokyo, Japan. This device provides an oscillometric record. The recorders employed in the current study had previously been validated and recommended for clinical use [19]. The monitoring equipment was arbitrarily applied at 8.00 A.M. The cuff was fixed to the non-dominant arm, and three blood pressure readings were taken concomitantly with sphygmomanometer measurements to ensure that the average of the two sets of values did not differ by >5 mmHg. All patients already taking antihypertensive drugs used their prescribed antihypertensive medications during ABPM, without changes in type, dosage or time of administration throughout the study. The device was set to measure blood pressure at 15-min intervals during the day (6 AM to 10 PM) and at 30-min intervals during the night (10 PM to 6 AM). During the 24 h of examination the patient was informed to hold the arm immobile at the time of measurements, to keep a diary of daily activities and to return to the hospital 24 h later. The monitoring was always done on a working day. The patients had no access to the ambulatory BP values.

Measurements recorded during the 24h were stored on a personal computer and screened as follows: a 24 h record was rejected for analysis if more than one third of potential day and night measurements were absent or invalid. The ambulatory BP values used for statistical analysis were expressed as 24-h average systolic and diastolic pressures, and 24-h average heart rate. Night/day ratio of BP was calculated as follows: mean nocturnal systolic BP/mean diurnal systolic BP, considering for the analysis the night/day BP ratio as a continuous variable. According to guidelines, [5] physiological nocturnal fall of blood pressure (dipper profile) is considered for a mean nocturnal reduction from 10 to 20% than mean systolic diurnal levels. Non-physiological nocturnal profiles are mild dipper (reduced nocturnal fall between 0 and 10%); extreme dipper (exaggerated nocturnal fall >20%) and reverse dipper (paradoxical nocturnal increase of BP). The indices of 24 h BP diurnal and nocturnal variability, were calculated for both systolic and diastolic BP as a standard deviation (SD) below the mean of the measurements for the period considered. We used the term “uncontrolled hypertension” referring to those cases of hypertension already known and under treatment in which the ABPM values obtained during the study remained above the normal threshold despite the therapy already used.

Statistical analysis of quantitative and qualitative data, including descriptive statistics, was performed for all items. Continuous data are expressed as mean ± SD, unless otherwise specified, categorical variables are expressed as percentage. A student T Test for unpaired data was used to compare the continuous variables between the two groups examined. The comparison of the proportions was performed using the Chi square test or the exact Fischer’s test (used when the expected frequency of the event was lower than five times). A multiple logistic regression analysis has been performed taking into account the role played by the main confounding factors (sex, age, diabetes, BMI, fasting glycemia) in influencing the relationship we found between the independent variable (IV) asthma and the dependent variable (DV) overall hypertension. We furthermore created a logistic regression model in which the DV was represented by the overall presence of hypertension in our asthmatic sample; the IV was represented by the use of increasing dose of ICS and was represented as a continuous variable. All P-values were two-sided and P-values less than 0.05 were considered statistically significant. Statistical analysis and graphs generation have been performed using R software version 4.0.2 (R Development Core Team—GNU General Public License—available free online: www.r-project.org)

## 3. Results

### 3.1. Anthropometric, Clinic and Laboratory Variables of Asthma Patients vs. the Control Group Free of Any Respiratory Disease

Table 1 shows the main anthropometric, clinic and laboratory variables analyzed both in asthma patients and in the control group free of any respiratory disease. Age of case group was 57.3 ± 12.7 years vs. 46.9 ± 12.3 years of controls (*p* < 0.001). The proportion of asthmatics also suffering from concomitant arterial hypertension was 75% (30 patients) vs. 45% (18 subjects) in the control group (*p*: 0.012); of these 12 asthmatics vs. 6 in the control group are newly diagnosed (*p*: 0.01) A significantly higher prevalence of type 2 diabetes mellitus (T2 DM) was found in asthmatics vs. controls (7 vs. 0 subjects, respectively; *p*: 0.006) without significant differences in fasting blood glucose levels (98.75 mg/dL vs. 93.72 mg/dL; *p*: 0.23). The BMI, an index associated both to T2DM development and to asthma severity, was significantly higher in the asthma group than in the control group (28.1% vs. 25.8%; *p*: 0.03).

### 3.2. ABPM Values in the Two Study Groups

Table 2 shows ABPM data of asthmatics vs. controls. We have observed, in asthma patients, higher levels of 24h-SBP (134.1 vs. 127.3 mmHg; *p*: 0.05) and a significantly higher morning blood pressure surges than in controls: 21.70 ± 12.8 vs. 12.0 ± 10.6 mmHg (*p* < 0.0001) for systolic blood pressure (SBP) and 18.0 ± 10.9 vs. 11.2 ± 10.0 mmHg (*p* < 0.005) for diastolic blood pressure (DBP).

Table 3 shows a comparison of nocturnal dipping patterns between asthmatics and controls, without finding statistically significant differences between the two groups.

### 3.3. Comparison Analysis of the Asthmatics with the Best FEV_1_ (Top Quartile) vs. Asthmatics with the Worst FEV_1_ (Bottom Quartile)

Table 4 shows demographic, clinic and lab variables of asthmatics comparing top vs. bottom FEV1 levels. Reduced FEV_1_ values are highly significantly associated with high blood pressure levels: 100% vs. 40% (*p* < 0.0001).

Table 5 shows ABPM data of asthmatics comparing top vs. bottom FEV_1_ levels. Reduced FEV_1_ values are significantly associated with higher 24 h-SBP levels: 142.25 ± 19.25 vs. 122.82 ± 15.50 mmHg (*p*: 0.005), higher 24h-DBP levels: 81.14 ± 8.36 vs. 73.26 ± 8.14 mmHg (*p*: 0.03), higher day SBP levels: 148.24 ± 19.51 vs. 128.48 ± 14.28 mmHg (*p*: 0.003), higher day DBP levels: 85.36 ± 8.99 vs. 77.48 ± 8.84 (*p*: 0.03).

Table 6 shows a comparison of nocturnal dipping patterns of asthmatics comparing top vs. bottom FEV1 levels. We have observed a trend among patients with better respiratory function to the association with the extreme dipper profile (*p*: 0.01).

### 3.4. Regression Analysis

Table 7 shows a multiple logistic regression analysis performed in order to verify the association between asthma, lung function parameters and hypertension taking into account the role played by the main confounding factors (age, sex, diabetes, fasting glucose and BMI for the association between asthma and overall hypertension and age, sex, severe asthma and asthma duration for the association between FEV_1_ % level and overall hypertension). The regression analysis confirms the independent role played by asthma with an odds ratio (OR) of 3.66 (CI: 1.29–11.1); *p*: 0.008, severe asthma (OR 4.32, CI: 1.88–9.54, *p*: <0.001) and FEV_1_% (OR 1.95, CI: 0.96–4.21, *p*: 0.01).

Figure 1 shows the results of the logistic regression model in which the dependent variable was represented by the overall presence of hypertension (also considering the newly diagnosed hypertension) in the asthma group and the independent variable was represented by the use of increasing dose of ICS, with no match in our case history of any positive relationship (*p*: 0.7193).

## 4. Discussion

The findings of our study strengthen the already well documented association between asthma and arterial hypertension. Our data show a prevalence of hypertension in asthmatics of 75%, up to 80.8% in patients with severe asthma; these values are significantly higher than that reported in other different epidemiological contexts [9,10,11] where ABPM was not used. The ABPM data, in addition to detecting a significant number of cases of hypertension not previously diagnosed, showed higher 24-h SBP levels in asthmatics than in controls. When compared with a control group with superimposable characteristics not affected by respiratory diseases, asthmatics also show a significantly higher BMI and higher prevalence of T2 DM. These findings confirm the association between overweight/obesity and asthma, although the rationale for this is not known [20]. Nevertheless these differences, the multiple logistic regression analysis confirms the close association between asthma and hypertension also taking into account the main confounding factors (sex, age, diabetes, BMI, fasting glycemia). In our sample the adjusted odds ratio (OR) for asthmatics to be also hypertensives is 3.66 (CI: 1.29–11.1) *p*: 0.008, higher than previously reported [8,9,10,11]. Moreover, it should be noted that even without considering the new cases diagnosed by ABPM, the prevalence of hypertension in asthmatics in our study (45%), tends to be higher than reported in the literature [8,9,10,11]. Our data therefore suggests that the prevalence of hypertension in asthma in Western populations may be higher than previously thought, and this data deserves further consideration. The analysis of the frequencies of the four 24-h nocturnal dipping profiles did not allow to link asthma to a specific nocturnal profile of BP. Despite this, interestingly, asthmatics with preserved lung function (top quartile for FEV1) showed a statistically significant different nocturnal BP profile than subjects with impaired lung function. The high frequency of the “extreme dipper” nocturnal blood pressure profile in patients with a relatively preserved respiratory function seems worthy of further evaluation, this being a pattern whose prognostic meaning is still under debate today.

Mechanisms underlying the high prevalence of arterial hypertension in asthmatics remain largely unknown. There are several theories trying to explain this epidemiological finding: reduced lung function (FEV1), impairment of smooth muscle tonus regulation, vascular remodeling, adverse effects of therapy, and genetic factors [16,21], which overlap with a background pattern of systemic inflammation. Chronic low-grade systemic inflammation is characterized by only moderate upregulation of circulating proinflammatory factors without clinical symptoms of inflammation [22,23]. The occurrence of inflammation as systemic rather than simply confined to the airways in asthma is a fascinating concept [24]. An inverse relationship between FEV_1_ % predicted and systemic levels of inflammatory markers was demonstrated [25,26], and systemic inflammation in the setting of chronic lung disease is a strong and consistent marker of future cardiovascular events [27,28]. Future studies are needed to assess the role played by systemic inflammation in the development of cardiovascular comorbidities in asthmatic patients; at present, available data concerning the association between asthma, systemic inflammation and development of atherosclerotic damage are largely inconclusive [29,30,31,32].

In our population reduced lung function, assessed by measurement of FEV_1_ levels, was the main feature associated to arterial hypertension. The findings of our study confirm previous observations on the association of low FEV_1_ levels with high BP [16]. In our population subjects belonging to the lowest quartile for FEV_1_ levels (mean value 45.2%) compared vs. the highest quartile (mean value 98.6%) reaches the statically significance for the association of hypertension (100% prevalent in low FEV_1_ quartile). The asthma patients in the lowest quartile for FEV_1_ also have higher 24-h and diurnal (but not nocturnal) SBP and DBP levels. To the best of our knowledge, this is the first observation of a peculiar association between reduced FEV_1_ and maintained physiologic nocturnal fall of BP, whereas preserved FEV_1_ seems to be featured by an exaggerated nocturnal decrease of BP (extreme dipper profile). This peculiar observation needs to be confirmed in a larger sample because of multiple determinants of the nocturnal BP fall [33,34,35]. On the other hand, a specific analysis performed classifying asthmatic subjects on the basis of the extent of the nocturnal BP fall do not allow us to find any association between a preserved (or lost) nocturnal reduction of BP and any of the parameters related to asthma severity, asthma duration or drug taken. Therefore, the clinical relevance of an impaired nocturnal BP profile in asthmatics need to be more deeply investigated.

Several authors have hypothesized that steroid therapy may play a role in the development of high BP in asthmatics. High corticosteroid doses has been associated with mineral corticosteroid effects [14] with retention of sodium and water, thereby predisposing to hypertension and atrial fibrillation [13,14] especially in subjects with a specific genetic background [36], but the cause-effect association between asthma therapy and the development of high BP values is still far from being demonstrated. Regarding the role of ICS therapy, Ferguson et al. [16], suggests a “double-edged sword” relationship with hypertension: lower doses of ICS may confer a “protective” association while the opposite may be applicable for higher doses. Our study do not demonstrate any relationship between the number of drugs taken daily or the increasing dose of ICS assumed and the probability of hypertension (see Figure 1). However, given the importance of the issue, it seems relevant to clarify if there is a threshold level of ICS therapy above which the benefit due to anti-inflammatory effect is overwhelmed by the sodium and water-retentive effects. There is also a need to establish definitively whether all corticosteroids have the same behavior or alternatively there are differences between the various compounds in relation to the different potency and/or the systemic absorption from the airways [15]. Anyway, the overall role and safety of long-term steroid therapy (both inhaled and oral) in asthma deserves an accurate assessment also in relation to the different profile of comorbidities of each patient.

Our study has some limitations, first of all the small number of the enrolled subjects that weakens the statistic power of our analysis, as such, these are preliminary reports of an ongoing study, whose aim is to increase the recruitment to investigate the unanswered questions. It should also be considered that we excluded patient with other concomitant respiratory diseases, such as rhinitis, COPD, OSA and/or bronchiectasis, which are also associated with low-grade systemic inflammation that could to different extent contribute to the risk of develop essential hypertension [24]. Moreover, we excluded active cigarette smoking patients; smoking is one of the main risk factors for high BP worldwide [5]. Taken together, it is plausible that the prevalence of hypertension in patients affected by overlapping respiratory conditions or who continue to smoke may be even higher in real life.

## 5. Conclusions

These preliminary results of the BADA Study support the idea that arterial hypertension in asthma is more prevalent than how supposed before, emphasizing the need for a deeper cardiovascular screening for asthma patients (mainly for those at higher cardiovascular risk). In particular, our data highlights the need for a more extensive use of ABPM, as recently stated also by other authors [37]; the 24-h monitoring carried out in all the patients enrolled made it possible to identify 12 newly diagnosed hypertensives (30% of the whole sample) among asthmatic patients.

Our data, although not conclusive, suggest the presence of specific alterations in the nocturnal BP profile associated with low FEV_1_ levels as well as the possible influence of medical therapy on the development of high BP values. The continuation of the BADA Study as well as other studies that may be undertaken in the future are called to clarify the still doubtful aspects of these issues.

## Figures and Tables

**Figure 1 ijerph-17-06925-f001:**
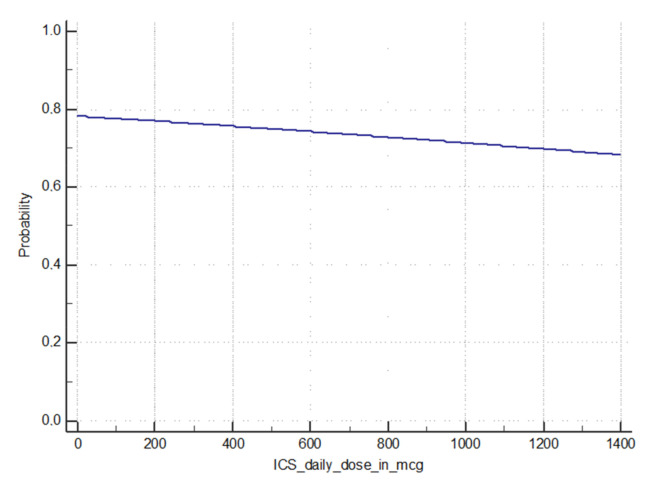
Logistic regression model—probability of hypertension in asthmatics with increasing daily intake of inhaled corticosteroid (ICS).

**Table 1 ijerph-17-06925-t001:** Demographic, clinic and lab variables—asthmatics vs. controls.

Variable	Asthmatics (n: 40)	Controls (n: 40)	*p*
M/F, n (%)	15/25 (37.5/62.5)	15/25 (37.5/62.5)	1
Age (years), mean ± SD	57.35 ± 12.67	46.87 ± 12.30	<0.001
Hypertension, n (%)	30 (75)	18 (45)	0.012
Newly-diagnosed hypertension, n (%)	12 (30)	6 (15)	0.01
Family history for hypertension, n (%)	25 (62.5)	25 (62.5)	0.8
Diabetes, n (%)	7 (17.5)	0 (0)	0.006
Fasting glucose (mg/dL), mean ± SD	98.75 ± 24.49	93.72 ± 9.49	0.23
Past cerebral vascular event, n (%)	0 (0)	0 (0)	-
Past cardiac vascular event, n (%)	0 (0)	1 (2.5)	-
Past peripheral arterial disease, n (%)	0 (0)	0 (0)	-
Creatinine (mg/dL), mean ± SD	0.83 ± 0.18	0.77 ± 0.18	0.6
Creatinine Clearance (mL/min), mean ± SD	96.36 ± 32.5	97.65 ± 16.80	1
Statin use, n (%)	6 (15)	2 (5)	0.13
BMI (Kg/m^2^), mean ± SD	28.10 ± 5.09	25.84 ± 4.86	0.03
Total Cholesterol (mg/dL), mean ± SD	190.81 ± 38.00	201.95 ± 44.39	0.8
HDL Cholesterol (mg/dL), mean ± SD	49.92 ± 17.68	57.95 ± 21.95	0.1
Triglycerides (mg/dL), mean ± SD	100.94 ± 38	112.19 ± 59.30	0.2
Current smokers, n (%)	0 (0)	0 (0)	-
Past smokers, n (%)	13 (32,5)	9 (22.5)	0.1
WBC (mm^3^)	8474.44 ± 3325	9181.90 ± 4450	0.3
Severe asthma, n (%)	26 (65)	-	
Asthma duration (years)	18.92 ± 14.77	-	
Oral steroid therapy, n (%)	4 (10)	-	
ICS low dose, n (%)	1 (2.5)	-	
ICS medium dose, n (%)	1(2.5)	-	
ICS/LABA low dose ICS (n/%)	5 (12.5)	-	
ICS/LABA medium dose ICS (n/%)	14 (35)	-	
ICS/LABA high dose ICS (n/%)	15 (37.5)	-	
Mean ICS daily dose (mcg)	485 ± 359.05	-	
LAMA (n/%)	16 (40)	-	
SABA (n/%)	23 (57.5)	-	
LTRA (n/%)	12 (30)	-	
Doxofylline (n/%)	2 (5)	-	
Biologic therapy (n/%)	5 (12.5)	-	
FEV_1_ (% predicted)	79.41 ± 23.27	95.10 ± 6.22	0.005
STEP GINA 2019 (n/%) 1,2 3,4 5	2 (5) 18 (45) 20 (50)	-	

Data are presented as mean value ± SD; BMI: body mass index; WBC: white blood cells; ICS: inhaled corticosteroid; LABA: long-acting beta agonist; LAMA: long-acting muscarinic antagonist; SABA: short-acting beta agonist; LTRA: leukotriene receptor antagonist; FEV_1_: forced expiratory volume in the first second. Creatinine clearance has been calculated through Cockroft and Gault formulae.

**Table 2 ijerph-17-06925-t002:** Ambulatory blood pressure monitoring (ABPM) data—asthmatics vs. controls.

Variable	Asthmatics (n: 40)	Controls (n: 40)	*p*
∆ D/N (%)	−13.34	−15.23	0.38
24-h SBP (mmHg)	134.15 ± 19.26	127.26 ± 12.79	0.05
24-h DBP (mmHg)	78.85 ± 10.44	78.72 ± 7.68	0.9
24-h HR (bpm)	76.36 ± 6.74	77.70 ± 8.11	0.15
Day SBP (mmHg)	139.29 ± 19.35	132.6 ± 12.24	0.07
Day DBP (mmHg)	82.67 ± 10.51	82.58 ± 7.62	0.9
Day HR (bpm)	79.15 ± 6.45	80.83 ± 8.74	0.3
Night SBP (mmHg)	120.71 ± 23.63	113.04 ± 16.02	0.09
Night DBP (mmHg)	67.44 ± 16.45	68.26 ± 9.76	0.8
Night HR (bpm)	69.02 ± 9.12	67.88 ± 8.97	0.56
Morning surge SBP (mmHg)	21.70 ± 12.8	12.0 ± 10.6	<0.0001
Morning surge DBP (mmHg)	18.0 ± 10.9	11.2 ± 10.0	0.005

Data are presented as mean value ± SD; ∆ D/N: percentage of reduction of nocturnal mean systolic blood pressure values in comparison to diurnal mean systolic blood pressure. SBP: systolic blood pressure; DBP: diastolic blood pressure; HR: heart rate; SD: standard deviation.

**Table 3 ijerph-17-06925-t003:** Nocturnal dipping patterns—asthmatics vs. controls.

Nocturnal BP Profile	Asthmatics (n: 40)	Controls (n: 40)	*p*
Dipper (n/%)	16 (40)	21 (52.5)	0.59
Mild dipper (n/%)	8 (20)	5 (12.5)	
Extreme (n/%)	12 (30)	12 (30)	
Reverse (n/%)	4 (10)	2 (5)	
Tot (n/%)	40 (100)	40 (100)	

BP: blood pressure.

**Table 4 ijerph-17-06925-t004:** Demographic, clinic and lab variables—asthmatics top vs. bottom levels of FEV_1._

Variable	Top (n: 10)	Bottom (n: 10)	*p*
FEV_1_ (% predicted)	98.6 ± 2.54	45.20 ± 14.11	<0.0001
M/F, n (%)	3/7 (30/70)	5/5 (50/50)	0.4
Age (years), mean ± SD	61.1 ± 17.20	58.5 ± 12.16	0.7
Hypertension, n (%)	4 (40)	10 (100)	<0.0001
Uncontrolled hypertension, n (%)	1 (25)	5 (50)	0.06
Family history for hypertension, n (%)	7 (70)	6 (60)	0.64
Diabetes, n (%)	2 (20)	2 (20)	1
Fasting glucose (mg/dL), mean ± SD	103.75 ± 27.55	104.71 ± 32.87	0.9
Past cerebral vascular event, n (%)	-	-	-
Past cardiac vascular event, n (%)	-	-	-
Past peripheral arterial disease, n (%)	-	-	-
Creatinine (mg/dL), mean ± SD	0.81 ± 0.06	0.92 ± 0.25	0.2
Creatinine Clearance (mL/min), mean ± SD	91.95 ± 25.39	52 ± 41.61	0.02
Statin use, n (%)	2 (20%)	-	0.15
BMI (Kg/m^2^), mean ± SD	28.00 ± 4.31	28.69 ± 4.94	0.7
Total cholesterol (mg/dL), mean ± SD	202.2 ± 36.92	207.83 ± 37.30	0.7
HDL cholesterol (mg/dL), mean ± SD	53.58 ± 18.8	64.5 ± 4.95	0.1
Triglycerides (mg/dL), mean ± SD	111.4 ± 46	74.66 ± 10	0.02
Current smokers, n (%)	-	-	-
Past smokers, n (%)	1 (10)	5 (50)	0.06
WBC (mm^3^)	7295 ± 2909	8555 ± 2297	0.3
Severe asthma, n (%)	4 (40)	9 (90)	0.02
Asthma duration (years)	14.7 ± 8.73	30.1 ± 20.03	0.04
Oral steroid therapy, n (%)	1 (10)	2 (20)	0.5
ICS low dose, n (%)	1 (10)	-	0.3
ICS medium dose, n (%)	-	-	-
ICS/LABA low dose ICS (n/%)	2 (20)	-	0.14
ICS/LABA medium dose ICS (n/%)	4 (40)	3 (30)	0.6
ICS/LABA high dose ICS (n/%)	3 (30)	4 (40)	0.6
Mean ICS daily dose (mcg)	400.4 ± 331.5	526.4 ± 487.8	0.5
LAMA (n/%)	3 (30)	7 (70)	0.08
SABA (n/%)	4 (40)	7 (70)	0.2
LTRA (n/%)	2 (20)	5 (50)	0.4
Doxofylline (n/%)	-	1 (10)	0.3
Biologic therapy (n/%)	-	4 (40)	0.03
STEP GINA 2019 (n/%) 1,2 3,4 5	1 (10) 6 (60) 3 (30)	- 3 (30) 7 (70)	0.3 0.2 0.2

Data are presented as mean value ± SD; FEV_1_: forced expiratory volume in the 1st second; BMI: body mass index; WBC: white blood cells; ICS: inhaled corticosteroid; LABA: long-acting beta agonist; LAMA: long-acting muscarinic antagonist; SABA: short-acting beta agonist; LTRA: leukotriene receptor antagonist; FEV_1_: forced expiratory volume in the 1st second. Creatinine clearance has been calculated through Cockroft and Gault formulae.

**Table 5 ijerph-17-06925-t005:** ABPM data—asthmatics-top vs. bottom levels of FEV_1._

Variable	Top (n:10)	Bottom (n:10)	*p*
∆ D/N (%)	−16.13 ± 9.12	−15.11 ± 7.24	0.8
24-h SBP (mmHg)	122.82 ± 15.50	142.25 ± 19.25	0.005
24-h DBP (mmHg)	73.26 ± 8.14	81.14 ± 8.36	0.03
24-h HR (bpm)	78.77 ± 4.21	76.02 ± 7.17	0.3
Day SBP (mmHg)	128.48 ± 14.28	148.24 ± 19.51	0.003
Day DBP (mmHg)	77.48 ± 8.84	85.36 ± 8.99	0.03
Day HR (bpm)	81.06 ± 3.52	79.66 ± 7.64	0.6
Night SBP (mmHg)	108.99 ± 20.87	125.92 ± 21.84	0.09
Night DBP (mmHg)	62.46 ± 8.42	70.11 ± 10.48	0.09
Night HR (bpm)	72.46 ± 8.04	66.28 ± 7.61	0.09
Morning surge SBP (mmHg)	34.7 ± 21.58	36 ± 16.73	0.8
Morning surge DBP (mmHg)	21.6 ±17.32	24.7 ± 17.10	0.7

Data are presented as mean value ± SD; FEV_1_: forced expiratory volume in the 1st second; ∆ D/N: percentage of reduction of nocturnal mean systolic blood pressure values in comparison to diurnal mean systolic blood pressure. SBP: systolic blood pressure; DBP: diastolic blood pressure; HR: heart rate; SD: standard deviation.

**Table 6 ijerph-17-06925-t006:** Nocturnal dipping patterns—asthmatics top vs. bottom levels of FEV_1._

Nocturnal BP Profile	Top (n:10)	Bottom (n:10)	*p*
Dipper (n/%)	4 (20)	5 (70)	0.3
Mild dipper (n/%)	1 (30)	4(10)	0.3
Extreme (n/%)	5 (50)	1 (20)	0.01
Reverse (n/%)	-	-	-
Tot (n/%)	10 (100)	10 (100)	

FEV_1_: forced expiratory volume in the 1st second; BP: blood pressure.

**Table 7 ijerph-17-06925-t007:** Multiple logistic regression analysis of association between asthma, lung function parameters and hypertension.

Variable	OR	CI	*p*
Asthma ^1^	3.66	1.29–11.1	0.008
Severe Asthma ^1^	4.32	1.88–9.54	<0.001
FEV_1_% level ^2^	1.95	0.96–4.21	0.01

^1^ Adjusted odd ratios from multivariate model including age, sex, diabetes, fasting glucose, and BMI; ^2^ adjusted odd ratio from multivariate model including age, sex, severe asthma, and asthma duration.

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
