# Peer review of "Prevalence of Arterial Hypertension and Characteristics of Nocturnal Blood Pressure Profile of Asthma Patients According to Therapy and Severity of the Disease: The BADA Study"

_ijerph, 2020, doi:10.3390/ijerph17186925_

Round 1

Reviewer 1 Report

In this manuscript Raimondo et al reported pilot results from the case-control BADA Study of asthmatics and controls, evaluating the prevalence and characteristics of arterial hypertension. The strengths of the study are that ambulatory blood pressure monitoring (ABPM) was conducted to evaluate blood pressure over one 24-hour period, and detailed clinical characteristics were collected which enabled examination of various cardiometabolic risk factors. Below are some comments that may help improve the manuscript.

Major comments

  1. Based on the design of the current analysis, the term “newly-onset hypertension” is probably not accurate since there does not appear to be follow-up of study participants yet (except for the 24hr ABPM). These cases were probably hypertensive before the study already, but was only detected with the use of ABPM. “Newly-diagnosed hypertension” is perhaps more appropriate as used elsewhere in the text.
  2. It would be interesting to see a breakdown of how many existing hypertensive cases there were and how many were undiagnosed (detected only through ABPM) before this study in both asthmatics and controls, which should be reported in the Results section. The prevalence of hypertension in asthmatics was reported to be 75% in asthmatics (45% if excluding the 12 newly-diagnosed hypertensive asthmatics), which is substantially higher than the cited literature in the Introduction. What do the authors think might be the reasons?
  3. Related to No. 2, the asthmatics in this study appeared to carry much worse cardiovascular risk factors than controls (diabetes, BMI, etc.). Are there systematic differences in patients referred to the two hospital divisions (Division of Respiratory Diseases where asthmatics were recruited vs. Division of Internal Medicine and Stroke Care where controls were recruited)? The potential of selection bias warrants discussion in this type of case-control study.
  4. The comparison between asthmatics with the best FEV1 (top quartile) and asthmatics with the worst FEV1 (bottom quartile) needs to be described in the Methods section.
  5. The regression analysis was described as to evaluate “the association between prevalence of hypertension and daily dose of Inhaled Corticosteroid (ICS) therapy in asthmatic patients”, while line 220 uses the term “incident hypertension” (this is only applicable for studies with follow-up; see No.1). Please clarify what exactly the outcome was here. Also, if the purpose of the regression model is not prediction, Figure 1 does not add good information. Instead, effect size and confidence interval associated with the exposure of interest should be reported along with p-value. Was this model adjusted for any potential confounding factors?
  6. Associations/tests reported in this manuscript were largely bivariate, while it is clear from Table 1 that there were differences in a number of factors comparing the asthmatics vs controls (in age, diabetic status, BMI, etc.) These could be confounding factors for the association between asthma status and hypertensive status, or lung function and hypertensive status. Regression models should at least be explored accounting for potential confounders. Otherwise, with just the crude analyses presented, the conclusions were not very well supported.

Minor comments

  1. Please use more consistently a point as decimal point (right now some are commas), and a comma as thousands separator (or just not use thousands separator). Please also use more consistent numbers of decimal places.
  2. Please include the confidence levels of CIs when citing others’ work.
  3. Line 49: by “underestimated” did the authors mean something along the line of “undiagnosed” or “unrecognized”?
  4. The evaluation of asthma severity was mentioned at line 119-120, but the definition of severe asthma needs to be more explicitly described.
  5. Line 157: “whether” should probably be “whereas” or “while”? Also t-test is just for continuous variables.
  6. Line 162: usually the version of statistical software is included.
  7. Table 1: what is “familiarity for hypertension”? Family history?
  8. What are the racial/ethnic characteristics of the study subjects?
  9. Table 4: what is the definition of “uncontrolled hypertension”?
  10. Line 271: “does” is redundant.

Author Response

We would like to thank you for the detailed review of our manuscript. We greatly appreciate the effort you made concerning your critique for the review of our study. We have accepted all your suggestions and revised the article according to them.

Major comments

Based on the design of the current analysis, the term “newly-onset hypertension” is probably not accurate since there does not appear to be follow-up of study participants yet (except for the 24hr ABPM). These cases were probably hypertensive before the study already, but was only detected with the use of ABPM. “Newly-diagnosed hypertension” is perhaps more appropriate as used elsewhere in the text.

            Thank you for your kind suggestion, the term “Newly-diagnosed hypertension” is now replaced in the text.

It would be interesting to see a breakdown of how many existing hypertensive cases there were and how many were undiagnosed (detected only through ABPM) before this study in both asthmatics and controls, which should be reported in the Results section. The prevalence of hypertension in asthmatics was reported to be 75% in asthmatics (45% if excluding the 12 newly-diagnosed hypertensive asthmatics), which is substantially higher than the cited literature in the Introduction. What do the authors think might be the reasons?

            This is certainly an interesting insight and we want to thank the reviewer for raising it. The available data regarding the prevalence of hypertension in asthmatics are very limited and the more severe the asthmatic disease the stronger the association with high blood pressure levels seems to be. We hypothesize that the prevalence of hypertension in asthma in Western countries is higher than previously thought, and we will continue the BADA Study with the aim of further testing this hypothesis.

As suggested we have added the prevalence of the new cases of hypertension diagnosed by performing ABPM in the results section and implemented the discussion about this issue.

Related to No. 2, the asthmatics in this study appeared to carry much worse cardiovascular risk factors than controls (diabetes, BMI, etc.). Are there systematic differences in patients referred to the two hospital divisions (Division of Respiratory Diseases where asthmatics were recruited vs. Division of Internal Medicine and Stroke Care where controls were recruited)? The potential of selection bias warrants discussion in this type of case-control study.

            The two hospital divisions (Division of Respiratory Diseases and Division of Internal Medicine and Stroke Care) where the recruitment was carried out are part of the same hospital, which serves the same population, so it is unlikely that there may be epidemiological differences in terms of different prevalence of cardiovascular risk factors.

The comparison between asthmatics with the best FEV1 (top quartile) and asthmatics with the worst FEV1 (bottom quartile) needs to be described in the Methods section.

            We have changed this in the methods section, as you suggested.

The regression analysis was described as to evaluate “the association between prevalence of hypertension and daily dose of Inhaled Corticosteroid (ICS) therapy in asthmatic patients”, while line 220 uses the term “incident hypertension” (this is only applicable for studies with follow-up; see No.1). Please clarify what exactly the outcome was here. Also, if the purpose of the regression model is not prediction, Figure 1 does not add good information. Instead, effect size and confidence interval associated with the exposure of interest should be reported along with p-value. Was this model adjusted for any potential confounding factors?

We would like to thank the reviewer because we realized that the explanation of the regression analysis we performed was unclear and partly contradictory. We have therefore adjusted this both in the methods section and in the results and in the discussion by clarifying that we tried to perform a case-related analysis. We applied a logistic regression model  in which the dependent variable (DV) was represented by the overall presence of hypertension (also considering the newly diagnosed hypertension) in our asthmatic sample, given the value 1 if the diagnosis of HTN was present and the value 0 if the diagnosis was absent; the independent variable (IV) was represented by the use of increasing dose of ICS and was represented as a continuous variable. We therefore aim to establish if the use of increasing dose of ICS could be linked to a higher probability to have diagnosis of hypertension founding that we couldn’t establish a positive relationship (p value 0.7189)

Sample size is a present limit of our work; this limits the assessment of confounding factor both by using logistic regression and by using Cochran-Mantel-Haenszel method to generate an odds ratio adjusted for confounding factors. According to Peduzzi et al. (1996) if we would took into account all the possible covariates (age, BMI and diabetes) we had to increase our sample to about 160.

Also the quality of the figure has been improved in the revised version

Associations/tests reported in this manuscript were largely bivariate, while it is clear from Table 1 that there were differences in a number of factors comparing the asthmatics vs controls (in age, diabetic status, BMI, etc.) These could be confounding factors for the association between asthma status and hypertensive status, or lung function and hypertensive status. Regression models should at least be explored accounting for potential confounders. Otherwise, with just the crude analyses presented, the conclusions were not very well supported.

According to the reviewer's suggestion, a further statistical analysis of the data was performed by an expert. A multiple logistic regression analysis has been carried out with the aim to further verify the association we found between asthma and hypertension taking into account in different models the role played by the main confounding factors (sex, age, diabetes, BMI, fasting glycemia, asthma severity and duration). The regression analysis confirms the independent role played by asthma (with an OR of 3.66), Severe Asthma (OR: 4.32) and FEV1% level (OR: 1.95). We have therefore modified the abstract, the methods section, the results, Table 6 and the discussion accordingly.

Note: in table 1 the fasting blood glucose result has been recalculated and it has been modified from the first draft of the paper.

Minor comments

            Please use more consistently a point as decimal point (right now some are commas), and a comma as thousands separator (or just not use thousands separator). Please also use more consistent numbers of decimal places

            Thank you for your kind suggestion, we have changed the paper accordingly.

Please include the confidence levels of CIs when citing others’ work.

Thank you for your kind suggestion, we have changed the paper accordingly. The CIs, when available, have been added.

Line 49: by “underestimated” did the authors mean something along the line of “undiagnosed” or “unrecognized”?

We agree with you. This is paraphrased.

The evaluation of asthma severity was mentioned at line 119-120, but the definition of severe asthma needs to be more explicitly described.

Thank you for your kind suggestion, we have now added a more precise version of the definition in the revised version

Line 157: “whether” should probably be “whereas” or “while”? Also t-test is just for continuous variables.

Thank you for your kind suggestion, we have now improved the description of the statistical test used.

Line 162: usually the version of statistical software is included.

We have added the version of R software used for the statistical analysis of our study (4.0.2).

Table 1: what is “familiarity for hypertension”? Family history?

This is a spelling error. we have now replaced the correct term “family history” in the tables

What are the racial/ethnic characteristics of the study subjects?

All subjects enrolled, both in asthma and in control group are white; now we have added this information when describing the study population

Table 4: what is the definition of “uncontrolled hypertension”?

We thank the reviewer for giving us the opportunity to discuss this point: the term "uncontrolled hypertension" refers to those cases of hypertension already known and under treatment in which the ABPM values obtained during the study remained above the normal threshold despite the therapy already used. We have now better explained this point in the methods section.

Line 271: “does” is redundant.

The term that can be found in the paper in line 271 of the original version, (line 288 in the revised version) is not "does" but "doses", referring to corticosteroid doses. No variation was therefore made to the manuscript.

We hope that we have successfully changed our manuscript according to your suggestions and that we have provided all the necessary explanations. We also hope that the manuscript now fulfills your criteria, and the Journal criteria for publication.

Reviewer 2 Report

The work entitled " Prevalence of Arterial Hypertension and Characteristics of Nocturnal Blood Pressure Profile of Asthma Patients According to Therapy and Severity of the Disease: the Bada Study " evaluates the prevalence and characteristics of arterial hypertension as a comorbidity of asthma. For that, they have measured several anthropometric, clinical and laboratory variables and they measure several hypertension-related parameters by using a 24h Ambulatory Blood Pressure Monitoring device. As the authors point out, the prevalence of hypertension in asthmatics has been previously reported. Therefore, a deeper screening of cardiovascular disease should be considered within patients with asthma.

In general, the manuscript is clearly written, the methodology used is correct and the results are clear. However, there are some concerns that must be addressed before the publication of this work in the present form:

  1. It is well-know the clear association of hypertension with age. However, the age of asthmatic patients is significatively higher than in healthy control population. This could be clearly affecting the main results of the article. The authors should adjust the analyses by age.
  2. It is very interesting the relationship of lung function parameters with blood pressure levels. However, the population number included in each group (N=10) is very small to take any conclusion, especially taking into account the underlying heterogeneity of both asthma and hypertension itself. As the authors point out these are preliminary results from a bigger study (BADA). I kindly suggest, therefore, to increase the N number in order to get a clear conclusion for the present article.
  3. Due to the present relationship between FEV1 and blood pressure, a regression analysis between lung function parameters and blood pressure parameters could be of interest.
  4. The background and methods section in the abstract must be improved. The authors repeat the aim in both sections and there is a lack of background.
  5. The Ethics approval for performing the study must be indicated in the methods section.

Author Response

We would like to thank you for your expert and meticulous review of our manuscript. Thank you very much for your positive opinion regarding our manuscript. We put a lot effort in this study and we appreciate your opinion very much.

  1. It is well-know the clear association of hypertension with age. However, the age of asthmatic patients is significatively higher than in healthy control population. This could be clearly affecting the main results of the article. The authors should adjust the analyses by age.

Thank you for your kind suggestion. In the revised version of the paper we added a multivariate logistic regression analysis taking into account the main possible confounding factors (age, type diabetes and Body Mass Index)

2.It is very interesting the relationship of lung function parameters with blood pressure levels. However, the population number included in each group (N=10) is very small to take any conclusion, especially taking into account the underlying heterogeneity of both asthma and hypertension itself. As the authors point out these are preliminary results from a bigger study (BADA). I kindly suggest, therefore, to increase the N number in order to get a clear conclusion for the present article.

We strongly agree with the reviewer. we confirm that this is a pilot study which is still in progress in order to reach a larger sample size. We also want to emphasize that it is clearly stated in the manuscript that the results need further validation in a more extensive sample. There are other studies in literature that have been considered worthy of publication even with a limited sample size and we hope that this may also be the case for ours.

3.Due to the present relationship between FEV1 and blood pressure, a regression analysis between lung function parameters and blood pressure parameters could be of interest.

According to the reviewer's suggestion, a further statistical analysis of the data was performed by an expert. A multiple logistic regression analysis has been carried out with the aim to further verify the association we found between asthma and hypertension taking into account in different models the role played by the main confounding factors (sex, age, diabetes, BMI, fasting glycemia, asthma severity and duration). The regression analysis confirms the independent role played by asthma, Severe Asthma and FEV1% level.

We have therefore modified the abstract, the methods section, the results, Table 6 and the discussion accordingly.

4.The background and methods section in the abstract must be improved. The authors repeat the aim in both sections and there is a lack of background.

Following your kind suggestion, we completely changed the background section of the abstract in the revised version

5.The Ethics approval for performing the study must be indicated in the methods section.

The approval of the local Ethical Committee has been added in the methods section, as requested.

We hope that we have successfully changed our manuscript according to your suggestions and that we have provided all the necessary explanations. We also hope that the manuscript now fulfills your criteria, and the Journal criteria for publication.

Reviewer 3 Report

In this manuscript, Domenico Di Raimondo, Gaia Musiari, and the co-authors evaluated the prevalence and characteristics of arterial hypertension in asthmatic patients. In general, the manuscript and forms are logically organized and well written in English. It is an excellent work with great clinical significance. It will be appreciated if the authors could further address the following concerns:

  1. Hypertension is an aging-associated disease, there is a big age gap between asthmatics and controls in this manuscript, please explain how to exclude this confounding factor.
  2. Blood pressure may change depending on the different phases or severity of hypertension, antihypertensive drugs, other concomitant diseases(eg, diabetes, hyperlipemia), lifestyle, and diet, please detail the clinical characteristics such as the meantime to diagnostic, antihypertensive drugs, antidiabetic/hyperlipemia medicines, lifestyle, and diet the patient population and controls. 
  3. Patients with the worst FEV1 have significantly higher asthma duration, higher hypertension prevalence, and worse kidney function than those with the best FEV1, please provide multiple past lung function measurement results in the asthmatics to better exclude the confounding factor of asthma duration.
  4. It would be better if the author can provide lung function assessments such as FEV1%  in the controls, to better explain the association of lung function and hypertension.

Author Response

We would like to thank you for your expert and careful evaluation of our manuscript. Thank you very much for your positive opinion regarding our paper: we appreciate your opinion very much.

  1. Hypertension is an aging-associated disease, there is a big age gap between asthmatics and controls in this manuscript, please explain how to exclude this confounding factor.

According to the reviewer's suggestion, a further statistical analysis of the data was performed by an expert. A multiple logistic regression analysis has been carried out with the aim to further verify the association we found between asthma and hypertension taking into account in different models the role played by the main confounding factors (sex, age, diabetes, BMI, fasting glycemia, asthma severity and duration). The regression analysis confirms the independent role played by asthma, Severe Asthma and FEV1% level.

We have therefore modified the abstract, the methods section, the results, Table 6 and the discussion accordingly.

  1. 2. Blood pressure may change depending on the different phases or severity of hypertension, antihypertensive drugs, other concomitant diseases (eg, diabetes, hyperlipemia), lifestyle, and diet, please detail the clinical characteristics such as the meantime to diagnostic, antihypertensive drugs, antidiabetic/hyperlipemia medicines, lifestyle, and diet the patient population and controls.

We thank the reviewer for deepening some epidemiological aspects related to the groups under study: unfortunately we have no information available on the hypoglycemic therapy possibly used in diabetics, while instead we have information on statins intake (data shown in Table 1), we know that all enrolled subjects, both cases and controls were sedentary, with various and regular dietary habits. The average duration of arterial hypertension was 5.3 years in asthmatics and 4.6 years in controls. Following the suggestion of the reviewer we have added in the section of the methods a paragraph in which it is indicated that " in relation to the available anamnestic information collected at the time of enrolment there were no significant differences between cases and controls regarding the characteristics of the diet and the level of physical activity (all subjects enrolled were sedentary).

  1. Patients with the worst FEV1 have significantly higher asthma duration, higher hypertension prevalence, and worse kidney function than those with the best FEV1, please provide multiple past lung function measurement results in the asthmatics to better exclude the confounding factor of asthma duration.

We understand and agree with the reviewer's observations but unfortunately we do not have multiple past measurements of FEV1% of all asthmatic patients; however, in our opinion, the lack of this data does not significantly affect our results and we try to explain why:

1 - the worst respiratory function (evidenced by the lowest level of FEV1%) is clearly related to the longer duration of the disease but not (in our case history) to a worse overall risk profile of being hypertensive (overlapping age, overlapping diabetes, overlapping renal function, overlapping BMI, etc.).

2 - the duration of asthma itself, disconnected from the consensual worsening of respiratory function, is not in our case history associated with a higher risk of hypertension. In fact, as indicated in the discussion of the article, the only element related to asthma that seems to be associated in our sample with a higher risk of hypertension is precisely the reduced FEV1 (analysis not shown revealed no significant association with the duration of hypertension, therapy used, alteration of the circadian rhythm of blood pressure).

  1. It would be better if the author can provide lung function assessments such as FEV1% in the controls, to better explain the association of lung function and hypertension.

We want to thank the reviewer for the suggestion. We have added in the revised version of Table 1 also for the control group the value of FEV1% obtained during the preliminary screening of patients at enrolment, the addition of this data further confirms our results.

We hope that we have successfully changed our manuscript according to your suggestions and that we have provided all the necessary explanations. We also hope that the manuscript now fulfills your criteria, and the Journal criteria for publication.

Reviewer 4 Report

In this manuscript titled, " Prevalence of Arterial Hypertension and  Characteristics of Nocturnal Blood Pressure Profile of Asthma Patients According to Therapy and Severity of the Disease: the Bada Study", Domenico Di Raimondo et al., authors aimed to evaluate the prevalence of the cardiovascular comorbidities of asthma and their impact on the course of the disease. Overall, the manuscript is written clearly.  However, the manuscript appears preliminary.

  1. In abstract and main text, the P value should be used in the same format. For example P=0.012.
  2. Authors should improve the quality of figure 1.
  3. More volunteer participants should be recruited.

Author Response

We would like to thank you for reviewing our manuscript. We greatly appreciate the effort you made concerning your critique for the review of our study. We have accepted all your suggestions and revised the article according to them.

    In abstract and main text, the P value should be used in the same format. For example P=0.012.

Thank you for your kind suggestion. Both in the abstract and in the main text a unique format for P-value has been used (ex. p: 0,05)

    Authors should improve the quality of figure 1.

According to your kind suggestion figure 1 was largely restyled

    More volunteer participants should be recruited.

Thank you for raising this issue. we want to emphasize that this is a pilot study which is still in progress in order to reach a larger sample size. We also want to highlight that it is clearly stated in the manuscript that the results need further validation in a more extensive sample. There are other studies in literature that have been considered worthy of publication even with a limited sample size and we hope that this may also be the case for ours.

We hope that we have successfully changed our manuscript according to your suggestions and that we have provided all the necessary explanations. We also hope that the manuscript now fulfills your criteria, and the Journal criteria for publication.

Round 2

Reviewer 1 Report

Thanks to the authors for addressing the comments. Apologies for misreading "doses" in the Discussion section. Below are a few additional minor points based on the revised version:

  1. Lines 28-9: I would add confidence interval and the covariates included in the model. To say "The regression analysis confirms the independent role played by asthma" just by itself is too bold a claim I think. There could be unmeasured confounding from e.g. lifestyle factors, or shared genetic susceptibility.
  2. The t-test conducted for Table 1 are most likely unpaired tests since the asthmatics and controls are not paired?
  3. Lines 172-4: I would also make it more clear what the dependent and independent variable of interest were for this model, similar to how lines 174-7 describes the model between ICS doses and hypertension.
  4. Line 189-90: regarding glycemic control, if the authors are citing the fasting glucose numbers in Table 1 these need to be updated.
  5. Section 3.4.1.: description of model adjustments in the footnote of Table 7 is not exactly the same as the text above.

Author Response

REVIEWER #1

We would like to thank the reviewer for the further comprehensive review of our manuscript. We have accepted all your suggestions and revised the article according to them.

Minor comments

            Lines 28-9: I would add confidence interval and the covariates included in the model. To say "The regression analysis confirms the independent role played by asthma" just by itself is too bold a claim I think. There could be unmeasured confounding from e.g. lifestyle factors, or shared genetic susceptibility.

            Thank you for your kind suggestion, we have improved the abstract of the paper accordingly. However, since we had to keep the maximum number of 200 words provided in the instructions for authors, we needed to make other small changes in the text.

The t-test conducted for Table 1 are most likely unpaired tests since the asthmatics and controls are not paired?

We agree with you. We have corrected the inaccuracy in the text

Lines 172-4: I would also make it more clear what the dependent and independent variable of interest were for this model, similar to how lines 174-7 describes the model between ICS doses and hypertension.

Thank you for your kind suggestion, we have now described in more detail the regression analysis in statistical analysis section.

Line 189-90: regarding glycemic control, if the authors are citing the fasting glucose numbers in Table 1 these need to be updated.

Thank you for the proper observation, we have now corrected the inaccuracy in the 3.1 section of the results.

Section 3.4.1.: description of model adjustments in the footnote of Table 7 is not exactly the same as the text above

Thank you again for the observation. This inaccuracy is due to the reason that multiple different regression models have been developed in the supplementary statistical analysis considering different set of variables. The description of the confounding factors evaluated in the different models is more detailed now.

We hope that we have successfully changed our manuscript according to your suggestions and that we have provided all the necessary explanations. We also hope that the manuscript now fulfills your criteria, and the Journal criteria for publication.

Reviewer 2 Report

Thank you for making the changes in the revised manuscript. In general, I think that the quality of the manuscript improves after the review. All the comments have been properly assessed and only have some minor comments:

  1. It is well-know the clear association of hypertension with age. However, the age of asthmatic patients is significatively higher than in healthy control population. This could be clearly affecting the main results of the article. The authors should adjust the analyses by age.
    1. A multivariate logistic regression analysis taking into account age as a possible confounding factor has been added to the manuscript.
  2. It is very interesting the relationship of lung function parameters with blood pressure levels. However, the population number included in each group (N=10) is very small to take any conclusion, especially taking into account the underlying heterogeneity of both asthma and hypertension itself. As the authors point out these are preliminary results from a bigger study (BADA). I kindly suggest, therefore, to increase the N number in order to get a clear conclusion for the present article.
    1. The authors clearly explain the preliminary nature of their results, but they are still relevant to be published in the present form. Thank you for the explanation.
  3. Due to the present relationship between FEV1 and blood pressure, a regression analysis between lung function parameters and blood pressure parameters could be of interest.
    1. This suggestion was performed and clearly improves the clinical relevance of the article.
  4. The background and methods section in the abstract must be improved. The authors repeat the aim in both sections and there is a lack of background.
    1. This suggestion was performed.
  5. The Ethics approval for performing the study must be indicated in the methods section.
    1. The ethics approval number has not been included in the methods section, please include it.

Other comments:

  1. The term White to refer people could be offensive for some collectives. I would rather suggest to change it to Caucasian.

Author Response

REVIEWER #2:

We would like to thank you for your further review of our manuscript and for judging the quality of the manuscript as improved after the recommended adjustments. We appreciate your opinion very much.

Minor comments

    It is well-know the clear association of hypertension with age. However, the age of asthmatic patients is significatively higher than in healthy control population. This could be clearly affecting the main results of the article. The authors should adjust the analyses by age.

        A multivariate logistic regression analysis taking into account age as a possible confounding factor has been added to the manuscript.

    It is very interesting the relationship of lung function parameters with blood pressure levels. However, the population number included in each group (N=10) is very small to take any conclusion, especially taking into account the underlying heterogeneity of both asthma and hypertension itself. As the authors point out these are preliminary results from a bigger study (BADA). I kindly suggest, therefore, to increase the N number in order to get a clear conclusion for the present article.

        The authors clearly explain the preliminary nature of their results, but they are still relevant to be published in the present form. Thank you for the explanation.

    Due to the present relationship between FEV1 and blood pressure, a regression analysis between lung function parameters and blood pressure parameters could be of interest.

        This suggestion was performed and clearly improves the clinical relevance of the article.

    The background and methods section in the abstract must be improved. The authors repeat the aim in both sections and there is a lack of background.

        This suggestion was performed.

    The Ethics approval for performing the study must be indicated in the methods section.

        The ethics approval number has not been included in the methods section, please include it.

Thank you for your further indication, we have added the ref number.

Other comments:

  1. The term White to refer people could be offensive for some collectives. I would rather suggest to change it to Caucasian.

Thank you for your kind suggestion, we have changed the term white into Caucasian, it was not the willingness of the authors to be offensive to any category of subjects.

We hope that we have successfully changed our manuscript according to your suggestions and that we have provided all the necessary explanations. We also hope that the manuscript now fulfills your criteria, and the Journal criteria for publication.